# Chemical Authentication and Speciation of *Salvia* Botanicals: An Investigation Utilizing GC/Q-ToF and Chemometrics

**DOI:** 10.3390/foods11142132

**Published:** 2022-07-19

**Authors:** Joseph Lee, Mei Wang, Jianping Zhao, Bharathi Avula, Amar G. Chittiboyina, Jing Li, Charles Wu, Ikhlas A. Khan

**Affiliations:** 1National Center for Natural Products Research, School of Pharmacy, University of Mississippi, University City, MS 38677, USA; 2Natural Products Utilization Research Unit, Agricultural Research Service, United States Department of Agriculture, University City, MS 38677, USA; 3Botanical Review Team, Office of New Drug Product, Office of Pharmaceutical Quality, Center for Drug Evaluation and Research, Food and Drug Administration, Silver Spring, MD 20993, USA; 4Division of Pharmacognosy, Department of BioMolecular Sciences, School of Pharmacy, University of Mississippi, University City, MS 38677, USA

**Keywords:** *Salvia* spp., GC/Q-ToF analysis, chemometrics, quality evaluation, chemical fingerprints

## Abstract

Members of the genus *Salvia* are used as culinary herbs and are prized for their purported medicinal attributes. Since physiological effects can vary widely between species of *Salvia*, it is of great importance to accurately identify botanical material to ensure safety for consumers. In the present study, an in-depth chemical investigation is performed utilizing GC/Q-ToF combined with chemometrics. Twenty-four authentic plant samples representing five commonly used *Salvia* species, *viz*. *S. apiana*, *S. divinorum*, *S. mellifera*, *S. miltiorrhiza*, and *S. officinalis*, are analyzed using a GC/Q-ToF technique. High-resolution spectral data are employed to construct a sample class prediction (SCP) model followed by principal component analysis (PCA) and partial least square discriminant analysis (PLS-DA). This model demonstrates 100% accuracy for both prediction and recognition abilities. Additionally, the marker compounds present in each species are identified. Furthermore, to reduce the time required and increase the confidence level for compound identification and the classification of different *Salvia* species, a personal compound database and library (PCDL) containing marker and characteristic compounds is constructed. By combining GC/Q-ToF, chemometrics, and PCDL, the unambiguous identification of *Salvia* botanicals is achieved. This high-throughput method can be utilized for species specificity and to probe the overall quality of various *Salvia*-based products.

## 1. Introduction

Members of the plant genus *Salvia* have a long and rich history of use as both culinary and medicinal herbs [1,2]. In general, the perennial shrubs have long stems which can reach heights of 50–100 cm. Although found throughout the world, most *Salvia* species grow in the Mediterranean region, Southeast Asia, and Central and South America [2]. Alluding to the importance of this plant’s medicinal properties, the word “*Salvia*” is derived from the Latin word “salvere”, meaning “to save” [2]. Members of the genus *Salvia* have been purported to possess a wide range of pharmacological properties, including anti-inflammatory, anti-dementia, anti-nociceptive, anti-hypertensive, anti-lipidemic, anti-mutagenic, anti-hyperglycemic, and anti-ischemic effects [1,2,3,4,5]. In addition to these purported properties, members of this genus have also been reported to possess anti-microbial and anti-oxidative activities [2,3,4,5]. These pharmacological properties vary among *Salvia* genus members.

Perhaps one of the most well-known members, *Salvia officinalis*, is utilized both as a culinary and a medicinal herb. This evergreen plant, native to southern Europe, is also cultivated in the United States and Central Asia [5]. As a medicinal herb, both the British Pharmacopoeia and the German Commission E have recognized its use to treat oral cavity and stomach ailments [5,6].

Another genus member, *Salvia apiana* (white sage), also has a rich history of use as a medicinal herb. This drought-resistant shrub, native to California and Baja California, can grow up to 1–3 m high. Traditionally, the herb has been used for its purported diuretic, anxiolytic, and anti-microbial properties [7]. In addition to its medicinal use, the plant is also an important part of traditional Native American religious and healing ceremonies [7].

*Salvia mellifera* (black sage), native to California and parts of Mexico, has also been used as a traditional healing herb. An infusion comprised of the aerial portions of the plant has traditionally been used as a drink to relieve muscle aches and pains [8].

As a popular ingredient in many traditional Chinese medicine (TCM) preparations, the red rhizomes of *Salvia miltiorrhiza* contain a unique group of compounds known as tanshinones, which have been reported to possess a broad spectrum of pharmacological activities [3,9]. The natural habitat of *S. miltiorrhiza* includes the hilly regions of China, Japan, Korea, and Mongolia; however, due to growing demand, most plant material is typically obtained from commercial farming [9]. Commonly referred to as “Danshen” or “Tanshen” in China, the rhizomes are purported to be beneficial for a number of disorders, including hyperlipidemia, vascular diseases, stroke, arthritis, and hepatitis [9].

*Salvia divinorum* is the only member of the *Salvia* genus to contain salvinorins, a group of neoclerodane diterpenes [10,11]. The plant, which can grow up to 1.5 m high, is native to southern Mexico. Traditionally, the Mazatec people would chew the herb or prepare an infusion using water and portions of the herb to take advantage of its psychoactive compounds. One compound, in particular, salvinorin A, possesses psychoactive properties and is a highly selective kappa-opioid receptor agonist [6,11]. Due to its high abuse potential, many local jurisdictions and countries have begun or are considering regulating the herb and/or salvinorin A as a controlled substance [6,11].

With nearly 900 species included in the genus *Salvia*, identifying plant materials and products can be a daunting task [4]. Thus, chemical fingerprint analyses represent a comprehensive approach for the quality assessment of *Salvia* botanicals and their finished products. Clearly, this is an important task given the wide range of pharmacological properties found in members of this genus. A range of methods have been developed to aid in the species identification of *Salvia* plant material [6,11,12,13,14,15,16,17]. Perhaps one of the most popular analytical techniques utilized for the identification and quality control of *Salvia* species is liquid chromatography/mass spectrometry (LC/MS) [6,12,13,15,16,17,18,19]. A brief literature search can yield numerous studies concerning this subject [6,12,13,15,16,17,18,19]. In addition to traditional LC/MS, techniques utilizing liquid chromatography/quadrupole time-of-flight (LC/Q-ToF) and liquid chromatography/triple quadrupole mass spectrometry (LC/TQ) have also been described [6]. While valuable to researchers, LC/MS/MS instruments are not often used by botanical industries for quality control purposes due to the cost of the instrument and the necessary technical skills required to develop and operate such tools. Due to the physical separation characteristics of LC/MS, the vast majority of previous research has focused on the non-volatile, LC-amendable polar compounds present in *Salvia* species [6,12,13,15,16,18,19]. Although polar compounds of *Salvia* are pharmacologically important, volatile constituents have also been implicated with bio-active properties and could be useful for establishing species-specific chemical fingerprinting [4,7,20,21].

DNA barcoding is another technique that has been proposed to aid in the species identification of *Salvia*. The authors of one study developed an effective DNA barcoding method to differentiate *S. miltiorrhiza* from other *Salvia* species [22]. Although differentiation was achieved, this technique involved extensive and complex sample preparation which did not lend itself to high-throughput sample analysis. In addition, the authors explained that their method was particularly developed for *S. miltiorrhiza* identification and may not be ideal for other *Salvia* species [22].

A high-performance thin-layer chromatography (HPTLC) fingerprinting method for 20 *Salvia* species was developed by Ciesla and co-workers [23]. The method utilized polar and semi-polar compounds (mostly polyphenols) for identification purposes and was successfully validated. Regarding its applicability as a high-throughput method, the authors estimated that 20 samples could be fully processed within one hour. However, one major limitation of this method was the requirement for a large amount of sample material (around 5 g), which could be problematic if the plant material is difficult to obtain, i.e., *S. divinorum* [23]. Therefore, there is a need to develop efficient and reliable methods.

Gas chromatography/mass spectrometry (GC/MS) is a well-established means of obtaining chemical fingerprints from various plants, primarily by analyzing volatile compounds. For example, this technique has been used to establish the chemical fingerprints of *Salvia* species by Rzepa and colleagues [24]. Based on the number of products sold in the U.S. market and pending botanical drug applications, five *Salvia* species, *viz*. *S. apiana*, *S. divinorum*, *S. mellifera*, *S. miltiorrhiza*, and *S. officinalis*, were selected to conduct a comprehensive and comparative study for quality evaluation and identification purposes. Currently, to the authors’ knowledge, a comparative study of the five selected *Salvia* species has not been conducted. Given these five species’ extensive history and current use as medicinal herbs, it is important to develop reliable and efficient identification methods for species specificity purposes and to assure the overall quality of various *Salvia*-based finished products.

With this information in mind, our goal is to develop a simple, reliable, and efficient GC method coupled with accurate mass spectrometry to establish species-specific chemical fingerprints of *Salvia*. Chemometric analysis and principal component analysis (PCA) are applied to differentiate between *Salvia* species, as well as to establish a sample class prediction model (SCP) based on partial least square discriminant analysis (PLS-DA) for the quality evaluation of commercial products. Marker and characteristic compounds present in each of the five species are identified. The integration of analytical data with statistical tools and the development of personal compound databases and libraries (PCDL) are anticipated to expedite the rapid evaluation of the quality of *Salvia*-based finished products, including raw materials used in commerce.

## 2. Materials and Methods

### 2.1. Plant Material

Twenty-four authentic plant samples from five *Salvia* species were used for this investigation. The samples included both leaf and aerial portions of the plants from *S. divinorum*, *S. officinalis*, *S. mellifera*, and *S. apiana*, with each having 3, 7, 3, and 5 individual samples, respectively. *S. divinorum* samples were procured from Trish Flaster (Botanical Liaisons, LLC, Boulder, CO, USA) (#578) and cultivated at the Medicinal Plant Garden (University, MS, USA) (#18434, #22491). *S. officinalis* samples were obtained from the Missouri Botanical Garden (St. Louis, MO, USA) (#7917, #7686, 20712), China (#16732), Richters.com (#13095), Trish Flaster (#2852), and Williams Warehouse (USA) (#1523). *S. mellifera* samples were sourced from AHP (Scotts Valley, CA, USA) (#22771, #22772), and SageRageHerb (Montclair, CA, USA) (#22506). *S. apiana* samples were obtained from AHP (#22773), Richters.com (#13096), SageRageHerb (Montclair, CA, USA) (#22502), and commercial sources (#22497, #22498). Six individual samples from the root portion of *S. miltiorrhiza* were also investigated. These samples were procured from Harvard Medical School (Boston, MA, USA) (#9729), the Medicinal Plant Garden (#11750), Beijing Yuke Botanical Development Co. Ltd. (Beijing, China) (#767), Missouri Botanical Garden (#8676, #12535), and a commercial source (#5399). The authenticity of the collected botanical samples was established based on morpho-anatomical and organoleptic properties by Dr. John Adams, a taxonomist at the National Center for Natural Products Research (NCNPR), University of Mississippi. In addition, DNA barcoding was also used for species verification purposes. Voucher samples of all the botanical material were deposited in the Botanical Repository of the NCNPR. The detailed sample information is given in Table 1.

### 2.2. Chemicals

Dichloromethane was purchased from Fisher Scientific (Pittsburgh, PA, USA). Both internal standards, tridecane (C_13_H_28_) and docosane (C_22_H_46_), were obtained from Polyscience Corporation (Niles, IL, USA). The reference standards, α-pinene, β-pinene, 3-carene, eucalyptol, camphor, endo-borneol, β-caryophyllene, viridiflorol, α-bisabolol, tanshinone II, cryptotanshinone, salvinorin A, and salvinorin B, used to confirm compound identification, were purchased from Sigma-Aldrich (St. Louis, MO, USA), Agilent Technologies, Inc. (Santa Clara, CA, USA), or isolated from plant material in-house at the NCNPR.

### 2.3. Sample Preparation

Dry, solid plant material from each species was ground and homogenized utilizing a ball mill. Approximately 100 mg of the powdered sample material was carefully weighed and placed into a small centrifuge tube. Samples for GC/Q-ToF analysis were prepared using a two-step method. Two internal standards (C_13_H_32_ and C_22_H_46_) were selected. Each standard was combined with dichloromethane to obtain a solution with a concentration of 100 µg/mL of each internal standard. First, 340 µL dichloromethane with 80 µL of the prepared internal standard solution was added to the samples and sonicated for 1 hour. Next, the samples were centrifuged for 10 min. This procedure was repeated one more time without adding the internal standards, after which the supernatant was collected and filtered prior to the GC/Q-ToF analysis. Each sample was prepared in duplicate.

### 2.4. GC/Q-ToF Analysis

All prepared samples were analyzed using an Agilent 7890B (GC) instrument equipped with an RS185 PAL3 autosampler. The GC was connected to an Agilent 7250 accurate-mass Q-ToF mass spectrometer. The capillary column (30 m × 0.25 mm i.d.) was coated with a 0.25 µm film of 5% phenyl methyl siloxane (J&W, HP-5MS). Helium at a constant flow rate of 1 mL/min was used as the carrier gas. Each sample was analyzed using the following GC oven program: 50 °C, held for 2 min, then heated at 2 °C/min to 280 °C, and finally held at 280 °C for 20 min. A post-runtime period of 5 min at 300 °C was also utilized. The inlet was programmed at 280 °C, while 1 µL of each sample was injected with a split ratio of 10:1. The transfer line from the GC to the Q-ToF was held at 300 °C. Duplicate injections were made for each sample.

The Q-ToF mass spectrometer was equipped with a high-emission low-energy electron ionization source which was operated with an electron energy of 70 eV and an emission current of 5.0 µA. During the experiment, the source, quadrupole, and transfer line temperatures were 280 °C, 150 °C, and 300 °C, respectively. All mass spectra data were recorded at a rate of 5 Hz from 35 to 500 *m/z* after a 5 min solvent delay. After every second sample injection, automated ToF mass calibration was performed utilizing a keyword command in the sequence table. Data were acquired utilizing Agilent MassHunter software (version B7.06.274, Agilent Technologies, Santa Clara, CA, USA). Further data processing was accomplished using Agilent MassHunter Qualitative Analysis and Quantitative Analysis (version 10.0.10305.0, Agilent Technologies, Santa Clara, CA, USA). The NIST database (version 2.3, NIST Standard Reference Materials, Gaithersburg, MD, USA) was utilized for tentative compound identification.

### 2.5. Data Processing and Statistical Analysis

As a part of data processing, the GC/Q-ToF data were converted into a .cef file format utilizing Agilent MassHunter Unknown Analysis (version 10.0.7070, Agilent Technologies, Santa Clara, CA, USA). The “SureMass” peak detection and deconvolution algorithm was elected, and a peak area filter of 10,000 counts was applied. Ions with identical elution profiles and similar spectral data were extracted as entities characterized by retention time (*t_R_*), peak intensity, and mass to charge ratio (*m*/*z)*. Then, the resulting .cef file for each sample was exported into the Mass Profiler Professional software package (version B.12.05, Agilent Technologies, Santa Clara, CA, USA) which includes SCP algorithms for further data processing.

After examining various minimum abundance counts, a setting of 5000 counts was finally selected for the extraction of entities from the spectra. The alignment of retention time, with a tolerance window of 0.15 min, and the similarity of the spectral pattern were carried out and compared across the entire sample set. The internal standard docosane (C_22_H_46_) was selected to normalize the peak intensity across all spectra. A stepwise reduction of entity dimensionality was performed based on common entities found across samples to further process the data. In addition, software settings such as parameter values (filter by flags), the frequency of occurrence (filter by frequency), the abundance of respective entities in classes (filter by sample variability), and one-way analysis of variance (ANOVA) were utilized and carried out by the software to filter the raw data. After filtering the raw data, quality control of the samples was performed by PCA to further reduce the dimensionality of the GC/Q-ToF data sets, increase interpretability, and minimize information loss. Based on the PCA, an SCP model was constructed. Five algorithms, namely, partial least squares discriminant analysis (PLS-DA), support vector machines (SVM), naive Bayes (NB), decision tree (DT), and neutral network (NN), were evaluated. The PLS-DA algorithm was selected since it was particularly well-suited for the project and resulted in the best prediction accuracy when compared to other algorithms. To validate the model, a k-fold cross-validation procedure was carried out. The validation procedure had three k-folds and was repeated ten times.

### 2.6. Establishment of a Personal Compound Database and Library (PCDL)

A PCDL was constructed using Agilent PCDL Manager software (version B8.00). Either readily available or isolated and fully characterized in-house chemical compounds were utilized as reference standards to establish the PCDL. Data including the retention time, exact mass, and high-resolution MS fragmentation patterns were exported to the PCDL. Additional information, such as the molecular formula, compound name, and CAS number were assigned to each entry for constructing the PCDL.

## 3. Results and Discussion

### 3.1. Extraction

Although hexane is touted as an ideal extraction solvent for capturing a wide variety of volatiles in botanicals, the extraction efficiency is questionable for some of the semi-volatile polar constituents, such as salvinorins from *S. divinorum*. These limitations with hexane are alleviated by utilizing dichloromethane [25] as the solvent of choice. A simple sample extraction procedure with dichloromethane improved the overall throughput and captured a wide variety of volatile analytes for species identification.

### 3.2. GC/Q-ToF Analysis 

After developing a satisfactory sample extraction technique and an optimized GC/Q-ToF method, the sample data were gathered (Figure 1). Upon examining the chromatograms of the investigated species, compounds were detected in the GC/Q-ToF analysis of the authentic *Salvia* plant extracts. Although there were slight variations among the concentrations of components within a particular *Salvia* species, characteristic and consistent fingerprinting patterns from the same species of *Salvia* were observed. However, distinct differences in their chemical profiles were noticed for different species, as illustrated in Figure 1.

Although approximately 200 compounds were tentatively identified from the five species, only 32 compounds which were found in the greatest abundance or were characteristic for each species were reported. The tentative identity of each analyte suggested by the NIST database was further confirmed with reference standards and the accurate mass of molecular ions when they were available for each analyte. Many early-eluting, highly volatile compounds were present in *S. officinalis*, *S. apiana*, and *S. mellifera*; however, these compounds were mostly absent in the samples of *S. divinorum* and *S miltiorrhiza*. After systematically examining the compounds present in each species, additional characteristic patterns were also established. For example, samples of *S. officinalis* contained the compounds β-thujone, viridiflorol, and verticiol, which were not detected in the other *Salvia* species. Although these compounds have been reported in other plant species, e.g., viridiflorol has been reported as a major constituent of *Allophylus edulis* [26], they were only present in *S. officinalis* among the five *Salvia* species in this study. Thus, the co-existence of β-thujone, viridiflorol, and verticiol can be used to distinguish *S. officinalis* from other *Salvia* species. This finding is also supported by a previous study comparing four *Salvia* species [27]. Likewise, samples of *S. mellifera* contained statistically significant amounts (*p* < 0.05) of camphor when compared to the other species. This is also consistent with Martino et al. report of *S. mellifera* containing approximately 12.2% camphor [28]. In addition, *S. mellifera* also contained β-amyrone, pectolinaringenin, and lupeol which were not detected in the other analyzed species. Only *S. apiana* samples contained γ-gurjunene and a statistically significant amount of isoledene. Unfortunately, due to the small amount of available literature concerning the volatile constituents of *S. apiana*, the authors were unable to confirm these findings with literature sources. *S. miltiorrhiza* samples contained the greatest amount (*p* < 0.05) of ferruginol, as well as the unique compound tanshinone II. The occurrence of tanshinone II in only *S. miltiorrhiza* samples is also supported by a review from Zhang et al. [9] In addition to being the only group that possessed the compounds salvinorin A and salvinorin B, *S. divinorum* also contained the greatest abundance (*p* < 0.05) of 8-hexadecyne. Willard and colleagues also reported the utility of salvinorin A in the identification of *S. divinorum* [25]. Utilizing these observed chemical distributions, each species’ chemical fingerprint and the peak area percentage of detectable compounds can be obtained (Table 2A,B).

### 3.3. Chemometric Analysis

Although the GC/MS identification of *Salvia* species is a popular means of species identification, it is often time-consuming [16]. While this method is well suited for small sample sizes, it does not lend itself to high-throughput applications, such as batch processing or quality control. With the coupling of GC to a Q-ToF mass spectrometer, vast amounts of high-resolution structural data can be gathered from compounds in each sample. Utilizing this data along with chemometrics, researchers can develop an SCP model from the data obtained from species [29,30].

PCA is a useful analysis that can transform large and complex data sets into manageable information for interpretation [31]. The stepwise reduction in entity dimensionality was performed based on filtering by flags, filtering by frequency, filtering by sample variability, and the results of ANOVA. Stepwise filtering intentionally created a strong filter so that the most discriminant entities could be used to construct the prediction model. After filtering, a PCA was performed, as illustrated in Figure 2. Good separation and species-specific clustering of the different *Salvia* species was achieved. Approximately 50% of the variation among species could be attributed to component **1**. Additional variation and separation could be explained by component **2** (15%). Contributing the least, component **3** only accounted for approximately 9% of the variation observed among the species.

Although the PCA demonstrated good separation between different *Salvia* species, it was unable to assign and predict the identity of unknown/commercial *Salvia* species sold in the U.S. market. Therefore, the GC/Q-ToF data for the authenticate samples were subjected to supervised chemometric methods. The first step in the SCP model construction process is to select the algorithm that is best suited to the project and the data set parameters. The PLS-DA [29] algorithm was found to be the best suited to construct a statistical model for *Salvia* classification and differentiation. Good separation obtained by the PLS-DA model among different *Salvia* species is shown in Figure 3. Once established, the software can use the sample characteristics and the associated algorithms to classify unknown samples. As illustrated in Figure 3, the PLS-DA successfully separated and clustered members of the authentic samples.

To validate the constructed model, the same authenticated samples used for the model training were repeatedly used due to the limited number of authenticated plant samples available. Although redundant, this is a valid statistical procedure (k-fold cross validation). Both the recognition and prediction abilities of the class prediction model were 100%, as shown in Table 3. Once the test was complete, a “confusion matrix” was generated. The test results indicated that this SCP could successfully identify and classify samples (Table 3). The construction of the SCP not only allows a large number of samples to be classified efficiently, but also in an automated manner. This allows the user to process additional samples at any point in the future.

### 3.4. Construction of a Personal Compound Database and Library (PCDL) for High-Throughput Screening

Although compound identification can be accomplished by manual inspection, this process can be both time-consuming and inefficient due to the large amount of high-resolution data obtained. With this in mind, a PCDL was constructed to facilitate the efficient throughput of samples. From the PCA loading plot (Appendix A), which is a visual representation of the “characteristic compounds” found in different *Salvia* species, marker compounds correlating to the separation of different species or the clustering of similar species were identified [30]. As illustrated in Table 4, each species could be distinguished by a few select compounds. Hence, the identified marker compounds that were commercially available or isolated in-house were analyzed by using the identical GC/Q-ToF method.

After analyzing the standards, data including the retention time, exact mass, and a curated accurate mass spectrum containing mass assignments for each spectral peak were exported to the PCDL. Utilizing the PCDL software, additional data such as the molecular formula, compound name, and CAS number were also captured. Figure 4 shows an overview of the PCDL table with the spectrum of salvinorin A, one of the marker compounds only present in *S. divinorum*.

The commercially available MassHunter Unknown Analysis software uses an algorithm called “SureMass” to find peaks in the accurate mass chromatogram and searches a mass spectral library or PCDL to identify compounds. If the library has locked retention times or index values, these can also be used as filters. If these filters are utilized, “hits” must have the correct retention time (*t_R_*) and be similar to the database spectrum. Figure 5 illustrates the results for the identification and isotope pattern for salvinorin A in one of the *S. divinorum* samples.

The “SureMass” peak-finding algorithm uses the added information available in high-resolution accurate mass data. For instance, extracted ion chromatograms of salvinorin A are overlaid and compared in Figure 5A. In contrast, a “head-to-tail” comparison plot of the high-resolution mass spectra of the suspected target and the reference compound illustrates the matching spectra (Figure 5B). In addition, the software can generate the compound’s isotope pattern if the molecular ion is detected in sufficient abundance. The compound’s theoretical value is next compared to the detected isotope’s *m*/*z* and relative abundance [30]. Additional confidence in the correct identification of the compound is provided when the theoretical value and detected *m/z* and abundance are good matches. In Figure 5C, the detected isotope pattern of salvinorin A (black vertical lines) is compared to the theoretical isotope pattern represented by red boxes. In the present study, peaks from the sample spectra of the five *Salvia* species that were identified by “SureMass” were compared to the in-house-constructed PCDL. This approach is inherently simple and data review is relatively easy. Once the PCDL is constructed, it not only allows for high sample throughput, but can be easily utilized in the future to analyze additional samples or be shared with research labs that do not have standard marker compounds.

## 4. Conclusions

Members of the genus *Salvia* have a rich history of both culinary and medicinal usage. With approximately 900 species included in the genus *Salvia*, the accurate species identification of processed botanical material can be a daunting task [2]. Although arduous, this task is of vast importance since the herb possesses species-specific pharmacological properties [2]. In the present study, we analyzed five species of botanically verified, medicinally important *Salvia* (*apiana, divinorum, mellifera, miltiorrhiza*, and *officinalis*) to develop a single analytical method for species differentiation purposes. Leveraging advances in software, the GC/Q-ToF of volatile organics, and the accurate mass spectral data allowed the unambiguous identification of five studied *Salvia* species. Although some of the marker compounds can be found in other plants, it is both the combination and concentration of the compounds that can aid in the species identification of *Salvia* botanical material. The implementation of chemometric analysis, *viz*. the PCA [29,30] of the *Salvia* samples, resulted in the identification of marker compounds for different *Salvia* species. Furthermore, the same PCA programs can also be expanded to build prediction models which may be utilized and modified for high-throughput sample analyses and classification purposes. To aid further, a PCDL combined with high-resolution mass spectrometry was developed with the versatility and ability to identify individual compounds present in *Salvia* samples.

In summary, by utilizing GC/Q-ToF, we obtained chemical fingerprints of each *Salvia* species being investigated. This information was further processed to construct an SCP model. By utilizing this model, future unknown samples can easily and efficiently be identified. As analytical needs change over time, the SCP model allows researchers to expand by including other economically important *Salvia* species. By leveraging advanced analytical techniques and chemometrics, the quality of closely related botanicals can be confirmed successfully, as demonstrated with a broad spectrum of biologically active *Salvia* species with complex chemistries.

## Figures and Tables

**Figure 1 foods-11-02132-f001:**
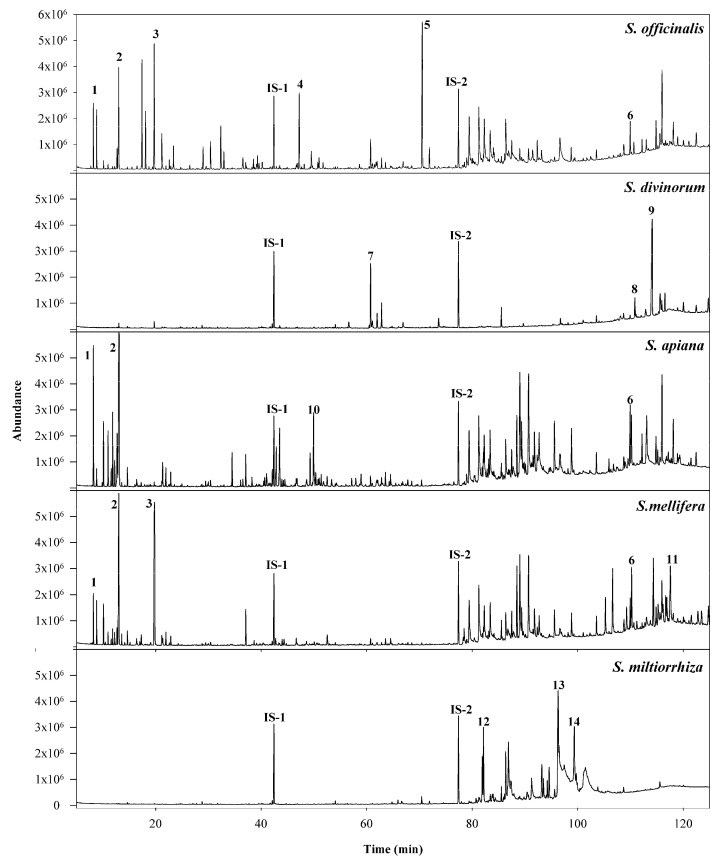
Representative chromatograms comparing *Salvia* species. Peak assignments: (1) α-pinene; (2) 1,8 cineole; (3) camphor; (4) viridiflorol; (5) verticiol; (6) salvigenin; (7) 8-hexadecyne; (8) salvinorin B; (9) salvinorin A; (10) γ-gurjunene; (11) lupeol; (12) ferruginol; (13) tanshinone II; (14) cryptotanshinone; (IS-1) tridecane; (IS-2) docosane.

**Figure 2 foods-11-02132-f002:**
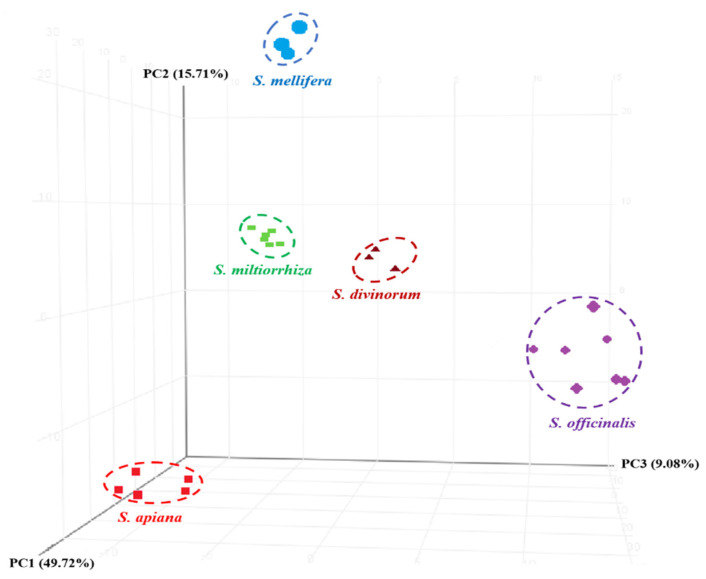
PCA score plot of five *Salvia* species.

**Figure 3 foods-11-02132-f003:**
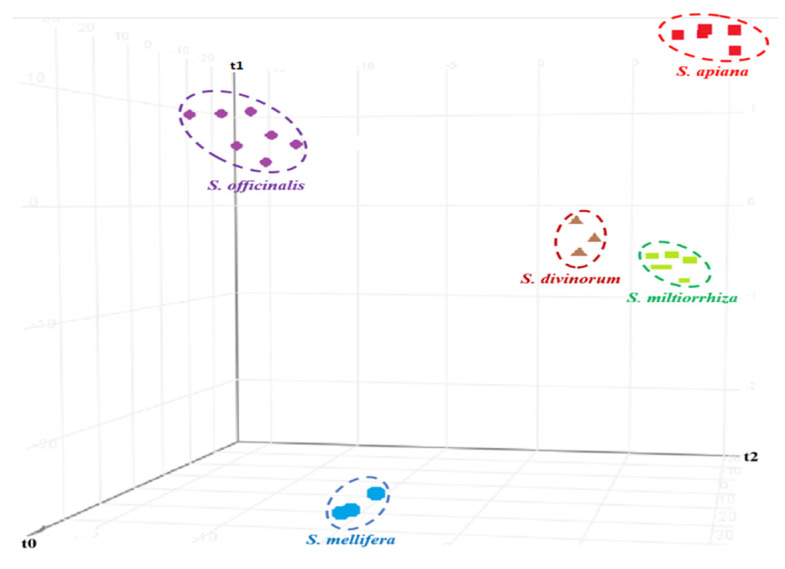
Score plot of the PLS-DA model constructed based on GC/Q-ToF data for the authenticated *Salvia* samples from five different species.

**Figure 4 foods-11-02132-f004:**
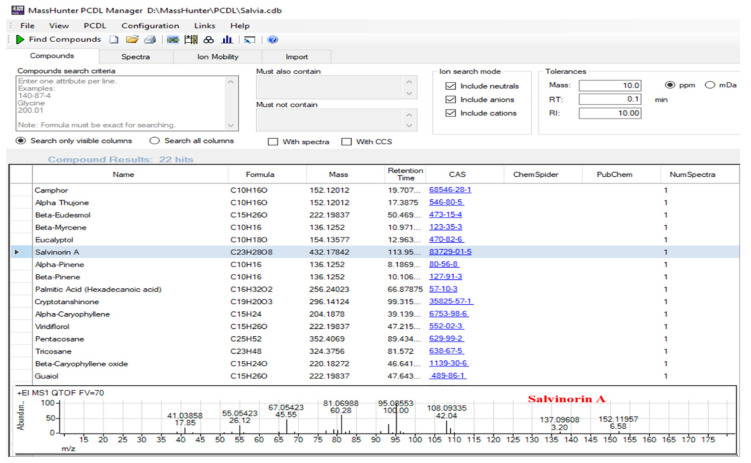
A section of the PCDL showing some of the content available for each entry and the accurate mass EI spectrum of salvinorin A from the PCDL.

**Figure 5 foods-11-02132-f005:**
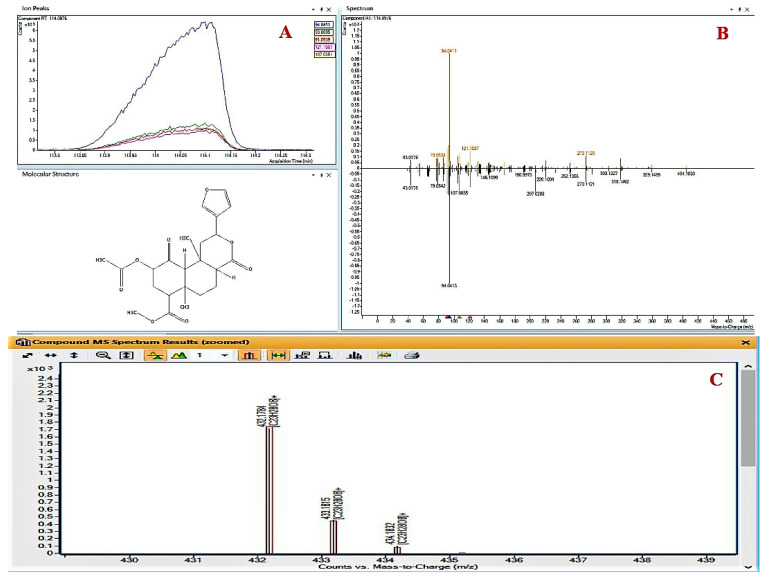
Identification of salvinorin A from *S. divinorum* (#22490). (**A**) Overlaid chromatograms of the five ions extracted for salvinorin A; (**B**) a “head-to-tail” comparison plot of high-resolution spectra of salvinorin A from PCDL (black) and the sample (orange); (**C**) the isotope pattern of the molecular ion (black vertical lines) compared to the theoretical pattern (red boxes).

**Table 1 foods-11-02132-t001:** Analyzed authenticated *Salvia* samples.

No.	NCNPR Code	Part	Botanical Name
1	1523	Leaf	*Salvia officinalis*
2	2852	Leaf	
3	7686	Mixed Parts	
4	7917	-	
5	13095	Leaf	
6	16732	Leaf	
7	20712	-	
8	13096	Leaf	*Salvia apiana*
9	22497	Aerial	
10	22498	Aerial	
11	22502	Aerial	
12	22773	Leaf	
13	578	Leaf	*Salvia divinorum*
14	18434	Aerial	
15	22491	Leaf	
16	22506	Aerial	*Salvia mellifera*
17	22771	Leaf	
18	22772	Leaf	
19	767	Root	*Salvia miltiorrhiza*
20	5399	Root	
21	8676	Root	
22	9729	Root	
23	11750	Root	
24	12535	Root	

**Table 2 foods-11-02132-t002:** Tentative compound identification based on NIST library and percent (% peak area) of volatile compounds in methylene chloride extracts of (A) *S. officinalis* and *S. apiana* and (B) *S. divinorum*, *S. mellifera*, and *S. miltiorrhiza* using GC/Q-ToF analysis.

A													
Compound	*t_R_* (min)	*S. officinalis*							*S. apiana*				
		1523	2852	7686	7917	13095	16732	20712	13096	22497	22498	22502	22773
α-Pinene ^a^	8.194	0.11	0.20	1.57	tr	1.91	0.54	tr	3.52	1.43	1.25	2.25	0.09
Camphene ^b^	8.817	0.29	0.20	1.99	0.15	1.66	0.48	tr	0.37	1.17	0.92	0.57	0.10
β-Pinene ^a,b^	10.113	tr	tr	1.99	tr	0.25	0.28	tr	1.47	1.32	1.38	1.90	0.10
3-Carene ^a^	11.852	nd	nd	nd	nd	nd	nd	nd	1.88	0.57	0.38	1.23	tr
Eucalyptol ^a,b^	12.983	3.14	2.11	4.34	1.36	4.41	1.16	1.47	10.90	6.20	5.89	9.59	3.04
α-Thujone	17.393	1.15	0.62	3.46	nd	nd	nd	nd	nd	nd	nd	nd	nd
β-Thujone ^b^	18.086	1.53	0.73	1.51	0.30	2.19	1.33	tr	nd	nd	nd	nd	nd
Camphor ^a^	19.697	3.73	5.32	6.33	3.61	6.18	3.40	4.32	tr	6.74	6.81	3.86	2.44
endo-Borneol ^a^	21.160	1.50	0.74	1.78	1.84	1.66	1.23	0.47	0.12	0.30	0.21	0.2	0.21
β-Caryophyllene ^a^	37.092	0.13	0.09	0.06	0.15	0.25	0.41	0.43	0.93	1.64	2.81	3.22	0.13
Isoledene ^b^	43.496	tr	tr	tr	tr	0.11	tr	0.23	2.35	1.84	2.59	1.05	0.33
Viridiflorol ^a^	47.199	2.73	3.14	5.38	3.93	3.54	4.14	1.94	nd	nd	nd	nd	nd
Humulenol ^b^	49.514	tr	0.15	1.94	0.98	0.77	0.93	1.71	nd	nd	nd	nd	nd
γ-Gurjunene ^b^	49.927	nd	nd	nd	nd	nd	nd	nd	2.73	1.46	2.06	1.01	0.23
α-Bisabolol ^a^	52.517	0.23	0.36	tr	nd	tr	nd	nd	0.28	0.41	0.66	2.59	0.33
8-Hexadecyne ^b^	60.731	0.11	1.32	0.59	2.86	1.22	1.63	4.15	0.28	0.31	0.41	0.43	0.75
3,7,11,15-Tetramethyl-2-hexadecen-1-ol ^b^	62.814	tr	0.47	0.24	1.16	0.42	0.53	1.26	0.21	0.21	0.35	0.36	0.32
Verticiol ^b^	70.524	8.19	12.17	7.81	10.59	9.75	10.17	3.84	nd	nd	nd	nd	nd
Aromandendrene ^b^	71.894	0.38	0.92	1.00	0.88	0.96	1.31	0.37	nd	nd	nd	nd	nd
Ferruginol ^b^	82.097	nd	nd	nd	nd	nd	nd	nd	0.36	0.07	0.09	1.15	0.48
Hexanedioic acid, mono(2-ethylhexyl)ester ^b^	85.533	0.65	2.88	2.83	0.96	0.25	0.67	1.24	0.42	0.44	0.50	0.45	0.49
Unknown	86.366	5.92	4.53	4.57	4.45	2.65	5.26	0.83	1.70	0.41	1.04	1.37	1.15
Salvicanol ^b^	88.476	nd	nd	nd	nd	nd	nd	nd	nd	1.41	1.19	2.05	2.45
Tanshinone II ^a^	96.184	nd	nd	nd	nd	nd	nd	nd	nd	nd	nd	nd	nd
Cryptotanshinone ^a^	99.365	nd	nd	nd	nd	nd	nd	nd	nd	nd	nd	nd	nd
Pectolinaringenin ^b^	105.420	nd	nd	nd	nd	nd	nd	nd	nd	nd	nd	nd	nd
Heptacosane ^b^	109.945	1.47	1.96	2.02	2.26	1.57	2.25	4.35	2.12	1.51	1.79	1.25	3.69
Salvigenin ^b^	110.291	0.60	0.06	0.05	0.06	0.64	0.64	0.29	1.72	4.26	3.45	2.01	5.90
Salvinorin B ^b^	110.841	nd	nd	nd	nd	nd	nd	nd	nd	nd	nd	nd	nd
Salvinorin A ^a^	114.026	nd	nd	nd	nd	nd	nd	nd	nd	nd	nd	nd	nd
β-Amyrone ^b^	116.668	nd	nd	nd	nd	nd	nd	nd	nd	nd	nd	nd	nd
Lupeol ^b^	117.441	nd	nd	nd	nd	nd	nd	nd	nd	nd	nd	nd	nd
**B**													
**Compound**	** *t_R_* ** **(min)**	** *S. divinorum* **			** *S. mellifera* **			** *S. miltiorrhiza* **					
		**578**	**18434**	**22490**	**22506**	**22771**	**22772**	**767**	**5399**	**8676**	**9729**	**11750**	**12535**
α-Pinene ^a^	8.194	nd	nd	nd	1.28	0.24	0.67	nd	nd	nd	nd	nd	nd
Camphene ^b^	8.817	nd	nd	nd	1.14	0.37	1.02	nd	nd	nd	nd	nd	nd
β-Pinene ^a,b^	10.113	nd	nd	nd	1.09	0.26	0.69	nd	nd	nd	nd	nd	nd
3-Carene ^a^	11.852	nd	nd	nd	0.38	tr	tr	nd	nd	nd	nd	nd	nd
Eucalyptol ^a,b^	12.983	1.12	0.31	1.13	6.44	3.55	4.20	0.67	0.81	tr	2.12	nd	1.67
α-Thujone	17.393	nd	nd	nd	nd	nd	nd	nd	nd	nd	nd	nd	nd
β-Thujone ^b^	18.086	nd	nd	nd	nd	nd	nd	nd	nd	nd	nd	nd	nd
Camphor	19.697	1.63	tr	3.71	10.21	8.75	9.73	0.40	2.13	1.95	0.90	0.05	0.84
endo-Borneol ^a^	21.160	nd	nd	nd	0.33	0.52	0.52	nd	nd	nd	nd	nd	nd
β-Caryophyllene ^a^	37.092	nd	nd	nd	1.28	0.73	0.82	nd	nd	nd	nd	nd	nd
Isoledene ^b^	43.496	nd	nd	nd	tr	0.13	0.11	nd	nd	nd	nd	nd	nd
Viridiflorol ^a^	47.199	nd	nd	nd	nd	nd	nd	nd	nd	nd	nd	nd	nd
Humulenol ^b^	49.514	nd	nd	nd	nd	nd	nd	nd	nd	nd	nd	nd	nd
γ-Gurjunene ^b^	49.927	nd	nd	nd	nd	nd	nd	nd	nd	nd	nd	nd	nd
α-Bisabolol ^a^	52.517	nd	nd	nd	0.50	1.70	0.69	nd	nd	nd	nd	nd	nd
8-Hexadecyne ^b^	60.731	12.09	10.45	9.32	0.22	0.50	0.45	nd	nd	nd	nd	nd	nd
3,7,11,15-Tetramethyl-2-hexadecen-1-ol ^b^	62.814	4.70	3.11	3.20	0.10	0.11	0.15	nd	nd	nd	nd	nd	nd
Verticiol ^b^	70.524	nd	nd	nd	nd	nd	nd	nd	nd	nd	nd	nd	nd
Aromandendrene ^b^	71.894	nd	nd	nd	nd	nd	nd	nd	nd	nd	nd	nd	nd
Ferruginol ^b^	82.097	nd	nd	nd	0.20	0.31	0.39	9.14	2.61	1.05	8.11	9.18	1.61
Hexanedioic acid, mono(2-ethylhexyl) ester ^b^	85.533	4.68	4.72	2.65	0.73	0.48	1.23	4.347	24.95	35.25	19.46	1.55	14.59
Unknown	86.366	nd	nd	nd	nd	nd	nd	nd	nd	nd	nd	nd	nd
Salvicanol ^b^	88.476	nd	nd	nd	3.23	2.55	2.81	nd	nd	nd	nd	nd	nd
Tanshinone II ^a^	96.184	nd	nd	nd	nd	nd	nd	15.25	13.19	12.70	16.91	12.57	12.14
Cryptotanshinone ^a^	99.365	nd	nd	nd	nd	nd	nd	nd	nd	nd	nd	12.16	nd
Pectolinaringenin ^b^	105.420	nd	nd	nd	1.63	4.22	3.98	nd	nd	nd	nd	nd	nd
Heptacosane ^b^	109.945	nd	nd	nd	2.55	1.93	2.21	nd	nd	nd	nd	nd	nd
Salvigenin ^b^	110.291	nd	nd	nd	2.55	1.93	2.21	nd	nd	nd	nd	nd	nd
Salvinorin B ^b^	110.841	3.60	0.75	1.12	nd	nd	nd	nd	nd	nd	nd	nd	nd
Salvinorin A ^a^	114.026	33.48	20.24	22.70	nd	nd	nd	nd	nd	nd	nd	nd	nd
β-Amyrone ^b^	116.668	nd	nd	nd	1.14	0.96	0.79	nd	nd	nd	nd	nd	nd
Lupeol ^b^	117.441	nd	nd	nd	4.16	3.34	4.09	nd	nd	nd	nd	nd	nd

nd: not detected; tr: trace amount; ^a^ compound identification based on NIST library was confirmed with reference standard; ^b^ accurate mass was consistent with GC/Q-ToF analysis.

**Table 3 foods-11-02132-t003:** Summary of classification results obtained by the PLS-DA model.

	*S. apiana*	*S. divinorum*	*S. mellifera*	*S. miltiorrhiza*	*S. officinalis*	Accuracy (%)
Model Training						
*S. apiana*	5	0	0	0	0	100
*S. divinorum*	0	3	0	0	0	100
*S. mellifera*	0	0	3	0	0	100
*S. miltiorrhiza*	0	0	0	6	0	100
*S. officinalis*	0	0	0	0	7	100
Recognition ability (%)	-	-	-	-	-	100
Model validation						
*S. apiana*	5	0	0	0	0	100
*S. divinorum*	0	3	0	0	0	100
*S. mellifera*	0	0	3	0	0	100
*S. miltiorrhiza*	0	0	0	6	0	100
*S. officinalis*	0	0	0	0	7	100
Prediction ability (%)	-	-	-	-	-	100

**Table 4 foods-11-02132-t004:** Proposed marker compounds tentatively identified for the differentiation of selected *Salvia* species.

No.	Compound ID	*t_R_* (min)	Formula	Base Peak	M^+^	Diff (ppm)	CAS Number
*S. officinalis*							
1	β-Thujone	18.086	C_10_H_16_O	67.0542	152.1196	0.22	471-15-8
2	Viridiflorol	47.199	C_15_H_26_O	105.0697	222.1975	−1.43	552-02-3
3	Verticiol	70.524	C_20_H_34_O	95.0853	290.2598	−2.13	70000-19-0
*S. divinorum*							
4	8-Hexadecyne *	60.731	C_16_H_30_	67.0542	222.2345	1.34	19781-86-3
5	Salvinorin B	110.800	C_21_H_26_O_7_	94.0413	390.1679	1.53	92545-30-7
6	Salvinorin A	114.026	C_23_H_28_O_8_	94.0412	432.1784	1.23	83729-01-5
*S. apiana*							
7	Isoledene *	43.496	C_15_H_24_	119.0854	204.1877	1.21	95910-36-4
8	γ-Gurjunene	49.927	C_15_H_24_	105.0702	204.1875	1.21	22567-17-5
*S. mellifera*							
10	Camphor *	19.697	C_10_H_16_O	95.0856	152.1195	−0.44	464-49-3
11	β-Amyrone	116.670	C_30_H_48_O	218.2034	424.3700	0.08	638-97-1
12	Lupeol	117.441	C_30_H_50_O	189.1639	426.3855	−0.28	545-47-1
13	Pectolinaringenin	105.42	C_17_H_14_O_6_	271.0607	314.0785	2.58	520-12-7
*S. miltiorrhiza*							
14	Ferruginol *	82.097	C_20_H_30_O	189.1275	286.2297	2.04	514-62-5
15	Tanshinone II	96.184	C_19_H_18_O_3_	261.0912	294.1252	0.52	568-72-9

* Statistically significant amount detected (*p* < 0.05).

## Data Availability

Data is contained within the article or Appendix A. The data presented in this study are available on request to the corresponding author.

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
