# Peer review of "Chemical Authentication and Speciation of *Salvia* Botanicals: An Investigation Utilizing GC/Q-ToF and Chemometrics"

_foods, 2022, doi:10.3390/foods11142132_

Round 1
Reviewer 1 Report
Paper title:
Chemical Authentication and Speciation of Salvia Botanicals: An Investigation Utilizing GC/Q-ToF and Chemometrics
Aim:
The aim of this research was to develop a simple, reliable, and efficient GC/MS-MS method to establish the species-specific chemical fingerprints. The chemometric analysis, such as principal component analysis (PCA) was applied to differentiate between Salvia species, and to establish a sample class prediction model (SCP) based on partial least square discriminant analysis (PLS-DA) for quality evaluation of commercial products. Furthermore, marker and characteristic compounds present in each of the five species were identified.
|
Section |
Review comments and notes |
|
Abstract, title and references |
The abstract is clearly written, with a good command of English and clear representation of the aim of the paper. Containing 220 words, it must be shortened in order to meet the technical demands of the journal Foods (200 max). Furthermore, it is adequately structured: background of the proposed research, both methods and multivariate statistical tool used, and main conclusions were mentioned. Line 21: Please, check double space. Line 25: Please, check double space.
The title of the paper adequately reflects the subject under investigation in the proposed study.
References are listed numerically, as demanded by formatting rules of the journal. Although there is no limitation in the number of references, a reference list of 26 citations is adequate for the topic proposed. Authors mainly use contemporary literature data – almost all references are published after year 2010, which is acceptable. |
|
Introduction/ background |
Throughout the introduction section, the authors show the awareness of the situation in the proposed field. However, in lines 118-126 the authors need to highlight why did they choose to analyze the 5 Salvia species described in this manuscript, among more than 900 known varieties. At the end of the introduction section a clear and concise aim of this study is given. Line 40: Please, check double space. |
|
Materials and Methods |
The materials and methods are well described: the plant material collection and authentication of the species, sample preparation, GC/MS analysis, and multivariate data analysis. Line 149: Please, check double space. Line 155: Please, check double space. Line 158: Please, check double space. Line 177: Please, check double space. Line 199: Please, check double space. Line 219: Please, check double space. Line 227: Please, check double space. Line 234: Please, check double space. |
|
Results |
The results report the successful applications of a GC/MS-MS system combined with PCA and PLS-DA data treatment. Lines 239-256: Please, align the paragraphs. Line 271: reference [23] should not be in italic. Lines 293-308: Please, align the paragraphs. Line 315: Please, check the double space. Lines 315-319: Which other models did the authors examine? Line 381: Please, check the double space. |
|
Discussion and Conclusions |
The meaning, relevance and importance of the obtained results are explained in the same section. They were adequately compared with previous studies. Conclusions are adequately supported by the results obtained. Lines 397 and 401: Please, check the double space. |
|
Overall |
The submitted research represents a novel apprach for verifying authenticity of celected 5 varieties of Salvia species, considering their importance as culinary and medicinal herb. However, the authors need to address and highlight the reason for choosing these 5 varieties among a vast number of varietes known to belong to Salvia species. |
Author Response
"Please see the attachment."

Reviewer 2 Report
The article investigated the applicability and performance of mass spectroscopy-paired gas chromatography (GC) combined with chemometric tools for the identification of species-specific marker compounds in sage products. Organizing the results into databases, then into libraries is of great importance for both the market and authentication purposes, since it allows rapid and more effective monitoring.
The Introduction basically provides a good summary of the relevant literature, the relevance of the topic, and correctly lists the objectives of the research. However, some modifications are suggested mostly in the “Materials and Methods” and “Results and Discussion” sections.
It is recommended to write the abbreviations in full once in the first place of publication (e. g., GC, MS, etc.). Generally, detail more the applied sample preparation and clarify the procedure of chemometric analyzes.
After the “Introduction”, no large spaces are used between paragraphs, please unify.
Table 1: It would be useful to add the origin of the samples to make it more transparent, and to indicate more clearly where samples (i. e., leaf, root, etc.) were taken for each species.
Line 166: Please the internal standards (C13H28 and C22H46).
2.3. Sample Preparation: Did you apply parallel sample preparation? There is no information about this.
2.4. GC/Q-ToF Analysis: Were the measurements repeated or was each sample analyzed only once? If the latter is the case, I am not sure that you have been able to construct sufficiently reliable classification models.
2.5. Data Processing and Statistical Analysis: Please extend the “quality control” part of this section. Why did you use them? How did you use them? What were the class variables? Did you validate the classification model? If yes, how did you do that?
Can you support your following statement with your own results: “A simple sample extraction procedure with dichloromethane improved the overall throughput and is expected to capture a wide variety of volatile analytes, which could be useful for species identification”?
Figure 1: How were these peaks determined and statistically evaluated, given that the sampling location (i. e., leaf, root, etc.) and the origin of the plant are not influential to analyses these chromatograms separately?
Table 2A and 2B: it would be more appropriate to refer to these tables as supplementary material and instead include a table summarizing statistically significant differences.
Lines 299-300: What exactly is SCP modelling here, please include in the relevant part of the "Materials and Methods".
Line 302: What do you mean by filtering the data?
Figure 2: Please also include the PCA loadings and explain the results, which components were significant for the separation. How influential were the different origins of the samples?
Figure 3: On what basis and from which other methods did the algorithm choose PLS-DA? How the internal validation was performed, in other words how the model selected the calibration and validation data sets? It is advisable to perform the calibration by omitting samples or sample groups according to criterion and then projecting the previously omitted data into the built model.
Table 3: By itself, this table does not carry extra information.
PLS-DA can also be used to identify components that function as markers, do you have results on this?
Lines 342-343: you are referring to PCA loadings. Are those results included in Table 4? Like the PCA loadings, the correlating variables (chemical compounds) can be determined for DA as well, and can be useful, inter alia, because they are the results of a supervised method.
How reliably can the identified components be used as markers, given the relatively small sample number and lack of parallel samples and measurement repetition? The findings are not really discussed and supported by former and/or recent researches… so this needs further amendment.
Please make the necessary additions, clarifications and amendments.
Author Response
"Please see the attachment."

Reviewer 3 Report
The reviewer appreciate the interest of the authors in the development of this manuscript. It is an interesting topic.
In this study, 24 plant samples representing five commonly used sage species were undertaken and analysed by GC/Q-ToF. The aim of the study was to establish species-specific fingerprints of sage and to construct a sample class prediction (SCP) model based on stepwise dimensionality reduction of the data. The data were subjected to principal component analysis (PCA) and partial least squares discriminant analysis (PLS-DA). The manuscript is generally well written. However, the following issues have to be addressed.
Line: 50-51. Please include the relevant citation of the British Pharmacopoeia and the German Commission E.
I suggest changing the word tea to infusion.
Line 87-88. What methods? Give examples with literature.
Line 89. Please this statement needs a bibliographic reference.
Line 96. “…previous research has focused on nonvolatile..” this statement needs a bibliographic reference.
Line 98. “…constituents have also been implicated with bio-active properties..” this statement needs a bibliographic reference.
Information on method validation is not provided (validation parameters).
Please extend the discussion - comparing the results obtained with the literature (Line 249-282).
The reviewer appreciate the interest of the authors in the development of this manuscript. It is an interesting topic. However, there is some not provided information. I suggest MAJOR REVISIONS.
Author Response
"Please see the attachment."

Round 2
Reviewer 2 Report
The article has been improved from the previous version, with the authors indicating where changes have been made. However, a few clarifications are still needed.
The materials and methods section should include what validation has been done (k-fold), the results were divided into how many groups during that, and which supervised methods have been used and selected as the most appropriate (SVM, etc.). The results section should only refer to this, not introduce it there for the first time.
You should indicate in the text that among the ~200 detected compounds 32 significant were included.
You should include in the “Materials and Methods” that you have applied various filtering on the data when you detail PCA.
I think this PCA loading plot representation rather strange. What do the axes cover and what components do the numbers in the plot cover? What are the “suggested marker compounds” in the figure (please name them)? Figure legend would be useful as well.
Again, please include in the “Materials and Methods” that you selected the algorithm that is best suited to the research. Did you select it based on classification accuracy?
After the necessary amendments have been made, the study can be considered for publication.
Reviewer 3 Report
The authors have completed the manuscript with the comments indicated by the reviewer. I recommend the paper for publication in the journal.
